# Keeping in Touch with Type-III Secretion System Effectors: Mass Spectrometry-Based Proteomics to Study Effector–Host Protein–Protein Interactions

**DOI:** 10.3390/ijms21186891

**Published:** 2020-09-19

**Authors:** Margaux De Meyer, Joren De Ryck, Sofie Goormachtig, Petra Van Damme

**Affiliations:** 1Department of Biochemistry and Microbiology, Ghent University, K. L. Ledeganckstraat 35, 9000 Ghent, Belgium; margaux.demeyer@ugent.be (M.D.M.); joren.deryck@psb.vib-ugent.be (J.D.R.); 2VIB Center for Medical Biotechnology, Technologiepark 75, 9052 Zwijnaarde, Belgium; 3VIB Center for Plant Systems Biology, Technologiepark 71, 9052 Zwijnaarde, Belgium; sofie.goormachtig@psb.vib-ugent.be; 4Department of Plant Biotechnology and Bioinformatics, Ghent University, Technologiepark 71, 9052 Zwijnaarde, Belgium

**Keywords:** AP-MS, BioID, effectors, infection biology, interactomics, type-III secretion, Virotrap

## Abstract

Manipulation of host cellular processes by translocated bacterial effectors is key to the success of bacterial pathogens and some symbionts. Therefore, a comprehensive understanding of effectors is of critical importance to understand infection biology. It has become increasingly clear that the identification of host protein targets contributes invaluable knowledge to the characterization of effector function during pathogenesis. Recent advances in mapping protein–protein interaction networks by means of mass spectrometry-based interactomics have enabled the identification of host targets at large-scale. In this review, we highlight mass spectrometry-driven proteomics strategies and recent advances to elucidate type-III secretion system effector–host protein–protein interactions. Furthermore, we highlight approaches for defining spatial and temporal effector–host interactions, and discuss possible avenues for studying natively delivered effectors in the context of infection. Overall, the knowledge gained when unravelling effector complexation with host factors will provide novel opportunities to control infectious disease outcomes.

## 1. Introduction

### 1.1. Effectors and Their Role in Establishing Host Interactions

For thousands of years, the coevolutionary arms race between bacteria and their hosts have provided opportunities for adaptations of bacteria–host associations. As a first line of defense, hosts have evolved to recognize molecular patterns of microbial origin, such as flagellin or lipopolysaccharide (LPS). These elicitors are termed either microbe-associated molecular patterns (MAMPs) or pathogen-associated molecular patterns (PAMPs). Since these terms are frequently used interchangeably, we will maintain the broader term MAMP [1]. MAMPs are detected by pattern recognition receptors (PRRs) that induce various antimicrobial and immune responses to eliminate encroaching infective agents, referred to as MAMP-triggered immunity (MTI) [2,3]. In turn, pathogenic and symbiotic bacteria have mastered powerful strategies to deceive these host immune surveillance systems to promote their own survival and dissemination. Part of their tactic consists of introducing a batch of virulence factors—termed “effectors”—into the host cell. The emergence of these host-injected bacterial effectors has in turn led to host responses recognizing and counteracting these effectors, leading to effector-triggered immunity (ETI), which often involves programmed cell death [2,4,5]. Although first discovered in plants, the ETI response has more recently been shown to also exist in mammalian cells [5]. Highly evolved pathogens, such as *Legionella*, *Salmonella, Pseudomonas*, and *Ralstonia* spp., or symbionts, such as *Rhizobium* spp., have further expanded their effector repertoire to suppress ETI or use it to their own benefit. For most of these bacteria, the injection of effectors has become essential for their association with the host. For instance, the acquisition of specific effector proteins by nonpathogenic *Escherichia coli* (*E. coli*) has led to the evolution of different “pathotypes”, such as enterohaemorrhagic *E. coli* (EHEC) and enteropathogenic *E. coli* (EPEC), pathotypes with high disease burden [6]. In a similar fashion, *Ralstonia solanacearum* mutants lacking the transcriptional activator HrpB, which controls expression of the type-III secretion system (T3SS) and effectors, show a reduced colonization, infection and multiplication ability *in planta* [7]. Strikingly, for certain pathogens such as *Chlamydiae*, up to 10% of the coding sequences encode secreted effectors, i.e., ~90 out of ~900 proteins, underpinning their importance in the pathogen’s life cycle [8].

### 1.2. Delivery of Protein Effectors by Specialized Secretion Systems

Although Gram-positives, such as *Listeria monocytogenes,* typically achieve effector release via general secretion pathways, in Gram-negatives, translocation of effectors into the host cell is facilitated by specialized evolved bacterial secretion systems. Nine different secretion systems have been discovered thus far [9], of which secretion system type-III, IV and VI (T3SS, T4SS and T6SS) provide a mean for the direct delivery of effector proteins into target cells that can be of either prokaryotic or eukaryotic origin. Many Gram-negative animal and plant pathogens (e.g., EHEC/EPEC, *Chlamydia*, and *Salmonella* spp., *Ralstonia* spp. and *Pseudomonas syringae* (*P. syringae*)) employ the T3SS, also referred to as “injectisome”. In recent years, exciting progress on the assembly, structure, and function of this molecular syringe has been made, as extensively reviewed in [10]. The T3SS was found to be evolutionarily related to the flagellar apparatus [11]. More specifically, this sophisticated 3.5 megadalton syringe-like system is composed of a cytosolic sorting complex that acts as the effector receiving component, several inner and outer membrane-embedded structures, a hollow needle spanning the three membranes upon host cell infection and the translocon complex at the tip (Figure 1) [12,13]. In contrast, the type IV secretion system (T4SS) is structurally related to the DNA conjugation system and can transport single proteins, protein complexes, or protein-DNA complexes into the prokaryotic or eukaryotic host cell. The type VI secretion system (T6SS) used by numerous Gram-negative bacterial species on the other hand, shares structural homology to phage tails and can, besides their primary function in antibacterial antagonism by means of secreted substrates, also secrete proteins into eukaryotic host cells. Gram-negative bacteria interacting with eukaryotic hosts most often harbor the T3SS. Although the technologies discussed in this review are in principle applicable to study effector–host protein–protein interactions (EH-PPIs) of all the above-mentioned secretion systems, we will mainly focus on type-III effectors (T3Es) and T3SS-related infection biology.

### 1.3. Effectors: Their Origin and Mechanisms of Action

As opposed to the highly conserved T3SS machinery [9], T3Es are functionally and structurally diverse [10]. As a safety net for adaptations in the host, a remarkably high degree of functional redundancy is common among T3Es, and is the product of continuous acquisition, modification and terminal reassortment of effector genes [11]. In terminal reassortment, new effectors arise as a result of genome rearrangements, e.g., the addition of an existing secretion domain to an existing gene [14]. Variations and adaptations to the effector repertoire of the pathogen or symbiont facilitate an intra- or extracellular lifestyle, distinct host ranges, and changes in pathogenicity. For instance, acquisition of large regions in the bacterial genome involved in pathogenicity, termed pathogenicity islands (PAIs), serve typical hallmarks of important host adaptations. This is illustrated by the establishment of an intracellular lifestyle of *Salmonella* upon acquisition of a second *Salmonella* pathogenicity island (SPI-2) through horizontal gene transfer in combination with an exquisite temporal expression control of both SPI-1 and -2 encoded proteins [15]. Furthermore, an in silico analysis of the genomes of multiple *Ralstonia solanacearum (R. solanacearum)* strains revealed that strains capable of infecting tomato plants possess two additional effectors (RipS3 and RipH3) compared to tomato nonpathogenic strains [16]. Indeed, the triple deletion mutant *ripH1-3* is significantly impaired in tomato pathogenicity [17], which demonstrates the involvement of effectors in shaping host range.

In contrast to bacterial proteins post-translationally transported through the general secretion (Sec) or twin-arginine translocation (Tat) dependent universal pathways, T3Es typically lack a conserved cleavable secretion signal [18], but possess an intrinsically disordered N-terminal ~20 amino acid long sequence enriched for serine, threonine, isoleucine and proline residues [19,20,21] and often a downstream 20 to 50 amino acid long chaperone-binding domain (CBD) [22]. Newly synthesized T3Es are generally believed to be kept in a partially unfolded state by chaperones that also target them to the T3SS sorting platform [10,23]. Only upon contact with the host, hierarchical secretion of the effectors by an ATP and proton-motive force-driven secretion mechanism through the T3SS is fully engaged. More specifically, at the sorting platform, effector proteins are stripped of their chaperones and further unfolded through ATPase activity, enabling their successive passage through the 20 Å hollow needle of the T3SS (Figure 1) [24]. Subsequent cytosolic delivery of the effectors in the host cell prompts folding into their active protein conformation. Although T3Es are thought to remain in an unfolded and thus inactive state inside bacteria, markedly, very recently, intra-bacterial enzymatic glycosylation activity of the *C. rodentium* non-LEE (locus of enterocyte effacement)-encoded effector NleB was demonstrated, which challenges the envisioned idea of T3Es residing dormant inside the pathogen [25].

Once inside the host cell, the effector operates by interacting with one or more host proteins. Diverse activities are ascribed to T3Es—sharing the common objective of modulating the host to the needs of the bacterium. In a controlled and integrated fashion, (sets of) effectors modify cytoskeletal dynamics, transcription, translation and signal transduction at multiple levels by covalent modification or binding of host protein targets, or by mimicking activities of host cell proteins among others. For instance, the phosphoinositide phosphatase activities of effectors IpgD (Inositol phosphate phosphatase D) and SopB (*Salmonella* outer protein B; also known as SigD) from *Shigella* and *Salmonella* spp., respectively, enzymatically modify plasma membrane phosphoinositides [26,27,28]. Alteration of the phosphoinositide composition of the plasma membrane causes membrane ruffling and actin cytoskeleton rearrangements in the host cell. Other representative examples include the actin-binding *Salmonella* effectors SipA/SspA (*Salmonella* invasion protein A/stringent starvation protein A) and SipC/SspC that reshape the actin cytoskeleton of the host by stimulating actin polymerization [29,30], and the phytopathogen *R. solanacearum* GalNAc-T activation (GALA) effectors that mimic plant E3 ubiquitin ligases to interact with SKP1-like proteins [31,32]. Also, in the context of beneficial interactions, effectors may play a key role. For example, the effector gene ErnA (effector required for nodulation-A) is conserved across many bradyrhizobia strains and triggers nodule formation in plants, as inoculation with the mutant results in loss of nodule formation. To properly achieve these functions, bacterial effectors are predicted to practice so-called “structural imperfect mimicry” in which a delicate balance between infectivity and toxicity has to be found [33]. By imperfectly simulating eukaryotic interaction interfaces, effectors can discriminate bacterial from host origin and steer their effects accordingly.

### 1.4. The Past and Future Ways of Studying Effector Interactomes

In the past few decades, examination EH-PPIs has largely relied on binary yeast two-hybrid (Y2H) screenings that entail the heterologous expression of a genetically tagged bait and prey protein in yeast, and using yeast survival as a readout [34]. The scalability and low cost of Y2H enabled semi high-throughput screening of prey libraries, such as the host target screens reported in the case of phytopathogens *P. syringae* and *Hyaloperonospora arabidopsidis* effectors probing a library of roughly 8000 *Arabidopsis* proteins [35] and for mapping of the EHEC host-pathogen interactome [36]. In the former study, Y2H screening showed that some host proteins, referred to as proteins hubs, often involved in immunity, are targeted by multiple effectors from evolutionary distant pathogens. Most of the interactions (33/51) found in Y2H between the effectors and the extensively targeted plant protein TCP14 were validated *in planta* [37]. To date, Y2H remains a popular screening platform for plant-based research, which can be coupled to next-generation sequencing (NGS), dubbed Y2H-sequencing (Y2H-seq), to further improve sensitivity and increase scalability [38]. Representative studies in which Y2H-seq was applied to study EH-PPIs include the identification of interactors of the HopZ2 (Hrp-dependent outer protein Z2) T3E from *P. syringae* implicated in plant host colonization [39] and the large-scale Y2H-screening for mapping interactions between effectors from the plant vascular pathogens *R. solanacearum* and *X. campestris* and *A. thaliana* protein [40]. However, high rates of false negative and positive interactions have been reported in Y2H-screens due to the non-physiological interaction conditions [41]. Recent advances in mass spectrometry (MS)-based proteomics strategies have unlocked opportunities for high-throughput and sensitive detection of EH-PPIs. In this review, we discuss the state-of-the-art MS-based toolbox for elucidating effector–host interactions and elaborate on the latest advances and shortcomings when studying EH-PPIs.

## 2. Methods to Elucidate Effector–Host Protein–Protein Interactions

### 2.1. Affinity- or Immune-Based Purification of Effector Interactomes in a Host Context

Reminiscent to Y2H being a predecessor of binary PPI approaches, affinity purification (AP) followed by MS (AP-MS) can be considered to be the founding MS-based approach for studying PPIs and protein co-complexation (reviewed in [42]). AP-MS consists of the expression of a translational fusion of the protein of interest (e.g., bait) to an affinity tag and consecutive co-purification of its interactors from native cell lysates by means of antibodies against the affinity tag (Figure 2), followed by proteolytic digestion of the sample (e.g., trypsin) which is subsequently subjected to MS [43]. By combining the separation capability of liquid chromatography (LC) with the analytic power of MS it has become a common practice to separate digests before sample injection into the mass spectrometer. By the online coupling of LC separation with MS, dubbed “LC-MS”, the overall sample complexity can be reduced, thereby providing a more comprehensive proteome coverage following MS-based analysis. AP can be done in a single or double step, depending on the selected (composite) affinity tag—of which diverse options exist, such as the FLAG-, HA- (i.e., hemagglutinin antigenic peptide), or tandem AP (TAP)-tag. In case of tagged bait expression, either high transfection efficiency models for overexpression of the fusion protein or, more ideal and gaining more interest in recent years, models expressing baits at endogenous levels can be used for carrying out AP-MS studies. In a similar fashion, immunopurification (IP) followed by MS (IP-MS) enables the co-purification of interactors of untagged baits by using antibodies specific for the selected bait protein. Unfortunately, weak and transient interactions are commonly missed in AP- and IP-MS (i.e., false negatives), viewing the loss of spatial organization upon cell lysis. Chemical crosslinking of proximal amino acids reactive side chains has emerged in recent years to surmount these problems [44], but has not been widely adopted in the field of effector–host interactions. Challenges with artefact integration, MS-spectra interpretation, and defining streamlined MS data analysis workflows likely accounts for the latter observation. False positive interactions, on the other hand, can be limited in AP-MS by use of appropriate reference controls, the use of an inducible expression system or the use of endogenous tagging. Another drawback intrinsic to the purification strategies used for AP- and IP-MS is the vast underrepresentation of membrane proteins, mainly because of the challenge to extract and solubilize these while maintaining physiological interactions during the purification process. Nevertheless, AP-MS has clearly proven of value for the identification of strong and less dynamic PPIs, or interactions enduring the applied lysis conditions, and the method is commonly used as reference method when asserting new PPI screening methods.

Classical AP-MS data only provides a static view of protein complexes. To facilitate the study of dynamic and regulated interactions, quantitative proteomics approaches have been implemented easing MS data interpretation and analysis. Metabolic protein labeling (e.g., stable isotope labeling of amino acids in cell culture (SILAC)) can be used to quantify relative differences in protein levels among light and heavy stable isotope-labeled samples (Figure 3) [45]. In case of SILAC, cells are differentially grown in medium containing either regular amino acids or amino acids (usually arginine or lysine) enriched with heavy carbon-13 (^13^C) and/or nitrogen-15 (^15^N) atoms, i.e., isotopologs. Consequently, the proteomes matching distinct experimental conditions are mass encoded when pooled, thus enabling the MS-based determination of (differences in) relative protein abundances (of interactors). When SILAC is coupled with IP, the protein lysates of the differentially SILAC-labeled cell populations undergo IP specific to the (tagged) bait protein (ectopically) expressed, added recombinantly, or—in case of effector biology—delivered in the context of a host-microbe interaction in a differentially SILAC labeled sample. In this way, relative differences in eluate protein levels can be determined by differential MS-based analysis, thereby pointing to potential interactors of the effector under study (Figure 3). SILAC is rarely used *in planta* due to the autotrophic nature of plants and thus poor metabolic labeling efficiency, although some efforts have been made (reviewed in [46]). Rather, plant researchers rely on post-metabolic labeling strategies or alternatively, label-free quantification (LFQ) for comparative quantitative proteomics.

Besides enabling the extensive mapping of protein complexes in various models in general [42] (e.g., the use of TAP in *Saccharomyces cerevisiae* [47]), AP-MS has also significantly contributed to the elucidation of EH-PPIs in bacterial infection biology (Table 1), such as in the large-scale identification of host interactors of *Chlamydia trachomatis* (*C. trachomatis*) vacuolar membrane protein (Inc) T3Es [48]. Mirrashidi et al. transiently expressed 58 Incs in human HEK293T cells and subjected them to large-scale AP-MS analysis, revealing 354 high-confidence Inc-host interactors [48]. Interestingly, the comprehensive EH-PPI dataset showed significant overlap with human host targets of viral proteins, suggesting shared pathogenic strategies for obligate intracellular microorganisms. This observation is further in line with a recent report in which a plant virus-encoded protein as well as the GALA1 and GALA3 *R. solanacearum* effectors convergently evolved to target chloroplasts, thereby interfering with salicylic acid (SA)-mediated defense [49]. Further research proved that IncE binding to sorting nexins (SNXs) 5/6 relocated them to the inclusion membrane. This observation was followed up by comparing the AP-MS interaction profiles of wild-type versus SNX5 mutated at the IncE binding surface, overall leading to the discovery that IncE disrupts a native interaction between cation-independent mannose-6-phosphate receptor (CI-MPR) and SNX5 and thus most likely represents an example of host mimicry [50]. Sontag et al. made use of AP-MS and came up with a collection of high-confident EH-PPIs of eight *Salmonella* and four *Citrobacter* effectors [51]. Interestingly, while for some *Salmonella enterica* serovar Typhimurium (*S*Tm) effectors, e.g., GtgA and SseI, 25, and 14 confident EH-PPIs, respectively, were retrieved, for four other effectors (CigR, PipB2, SifA, and SssA) no host interactors were found. In contrast, D’Costa and colleagues screened five *Salmonella* T3Es, including PipB2 and SifA, for human protein interactors using AP-MS in a cellular context (i.e., by the use stable epithelial cell lines and inducible effector expression), and identified a total of 130 putative EH-PPIs (including 63 EH-PPIs for PipB2 and 13 EH-PPIs for SifA) [52]. The observation that SifA and PipB2 localizations are steered by specific protein targeting mechanisms such as prenylation [53] and lipid raft-recruitment [54], respectively might (partially) account for the success of identifying EH-PPIs of these effectors in a cellular context. Recently, 9 and 12 *S*Tm effectors were screened using (crosslinking) AP-MS upon infection with chromosomally tagged effector strains in epithelial and macrophage cell lines, respectively [55]. In this study, Walch et al. described 412 novel candidate EH-PPIs, next to 25 previously observed interactors, of which 4 SifA and 16 PipB2 EH-PPIs were in accordance with the study of D’Costa and colleagues. Remarkably, only 28 EH-PPIs were identified in both cell lines studied, putatively evincing the different infection strategies used by *S*Tm in epithelial cells and macrophages in addition to the differential host protein expression profiles of both cell lines. Furthermore, several EH-PPIs, such as SifA with VPS39 and RBM10, were uniquely found after crosslinking before pulldown, suggestive of transient or weak EH-PPIs. In plants, several AP-MS-based studies investigating EH-PPIs have been performed [56,57,58,59]. A commonly used practice is GFP-trapping in which a translational fusion between the protein of interest and GFP is used to coprecipitate interacting proteins using beads coupled to an anti-GFP antibody. Using this technique, Sang et al. showed that the *R. solanacearum* T3E RipAY targets thioredoxins, proteins involved in redox regulation, of tobacco and Arabidopsis [59]. Furthermore, Üstün et al. showed that the *P. syringae* T3E HopM1 interacts with different proteasomal proteins of tobacco, including ECM29, which, interestingly, is involved in halting proteasomal protein degradation [59,60].

Various examples exist in which SILAC coupled with IP-MS was capable of pinpointing EH-PPIs upon mammalian host cell T3E expression or addition of the recombinant T3E to cell lysates [61,62,63,64,65,66,67]. This way, Shames et al. described the function of EspZ and NleC EPEC effectors by unravelling their interaction with host target CD98 and p300, respectively [62,63]. Direct interactions of both effectors to their corresponding host prey were confirmed by co-IP. Another study investigated the interaction partners of SPI-2 effectors in mammalian cells [64] and identified previously reported binding partners for SopB, SseJ (*Salmonella* secreted effector protein J), and SspH1 (*Salmonella* secreted protein H1). Besides, several new interactions were presented, such as junction plakoglobin and desmoplakin as host target of the less characterized effectors SseF and SseG, respectively. Furthermore, EH-PPIs for SseL with oxysterol-binding protein 1 (OSBP) and SspH2 with SUGT1 (Suppressor of G2 allele of SKP1 homolog) were verified. By means of co-IP and in the context of a stable inducible SopA-expressing cell line as well as *Salmonella*-infected cells, Fiskin and co-workers validated the interaction between SopA and two significant SopA-enriched hits, namely tripartite motif containing 56 (TRIM56) and TRIM65 [65].

### 2.2. Proximity-Dependent Labeling Approaches

In contrast to the previously discussed methods, proximity-dependent labeling approaches also capture proximal proteins besides direct and indirect PPIs. Use of a promiscuous biotin ligase (PBL) or a peroxidase (e.g., APEX (ascorbate peroxidase)) fused to a bait of interest enables covalent biotin labeling of vicinal proteins, and thus does not require the interaction of the protein complex to be maintained during purification or isolation, the latter being a major improvement considering that the use of different stringencies of (washing) buffers to extract proteins and remove nonspecific binders in AP-MS can heavily impact the proteins retained when performing affinity purifications. Following protein digestion, biotinylated proteins can subsequently be identified by means of LC-MS/MS. In proximity-dependent biotinylation (PDB; Figure 2), identification of protein interactors and proximal proteins is based on the enzymatic action of a PBL, an approach referred to as BioID when making use of mutant *E. coli* BirA^R118G^, designated BirA* (R118G), as PBL [68]. More specifically, upon supplementation of biotin to cells expressing a BioID fusion protein, the PBL moiety transfers activated biotin, or biotinoyl-AMP, to accessible primary amines in its vicinity [69]. This enables subsequent streptavidin-based purification and MS-based identification, overall permitting the identification of high as well as low affinity interactions. The robust biotin-streptavidin interaction permits the use of stringent washes (e.g., high content of (chaotropic) salts and detergents) and thus results in a reduced background of nonspecific proteins bound to affinity resins—providing significant improvements over AP-MS. Additionally, the ability for selective capture makes the method generally insensitive to protein solubility or protein complexation, and thus improves its applicability for interactome analysis of membrane proteins and cytoskeletal constituents, representing major advantages over alternative interactomics approaches. Occasionally, the relatively large BioID tag (35.3 kDa) may affect subcellular targeting of certain bait proteins. To steer its observed cytosolic localization in eukaryotic cells, BirA* can be engineered with a signal leader peptide to translocate the fusion protein to the desired compartment [70,71]. To potentially improve on these aspects, Kim and colleagues developed a smaller (27 kDa) second-generation PBL, called “BioID2”, from the *Aquifex aeolicus* biotin ligase, naturally lacking a DNA-binding domain [72]. Likewise, the Khavari lab engineered a mutant biotin ligase from *Bacillus subtilis* termed BASU (28 kDa) [73]. More challenging, however, is the relatively slow kinetics of these first- and second-generation enzymes used for PDB as sufficient biotinylation typically requires up to 16 to 24 h of labeling, overall complicating the exploration of dynamic interactions. To tackle the latter inadequacy, third-generation PBLs were developed by the directed evolution of BioID in yeast, namely TurboID (35 kDa) and miniTurbo (28 kDa) [74]. Alternatively, in vitro BioID (ivBioID) was shown to capture dynamic interactions by PBL labeling of permeabilized fixed cells, obtaining the cellular context in a physiologically inactive state and thus partially overcoming the limitation of slow BirA* kinetics while additionally increasing the spatial resolution [75].

BioID made its first introduction *in planta* in 2017 where researchers used rice protoplasts to investigate the vicinal proteins of OsFD2, a rice transcription factor, tagged with BirA* [76]. In 2018, Khan et al. applied the technique for identifying EH-PPIs in a whole plant system (transgenic *A. thaliana*) by searching for interactors of the *P. syringae* T3E membrane-associated HopF2 (Table 2) [77]. Interestingly, when compared to acquired AP-MS data showing the interaction of HopF2-HA with 10 membrane-associated proteins [56], nine extra membrane-associated prey proteins could be identified with BioID (i.e., 18/19 of the proximal proteins identified localized to membranes), thereby revealing some previously unknown targets of HopF2. Although 58% (11/19) of the strong HopF2 interactors identified by BioID were also identified with AP-MS, a few known interactors of HopF2 (MKK5, BAK1) were missed with BioID, a finding likely attributed to the non-accessible nature of free amines on proximal proteins for biotinylation, an observation in line with other reports [72,78]. Using BioID in tobacco leaves, researchers identified five proximal plant proteins to interact with the *P. syringae* T3E AvrPto, including RIN4 which functions at the membrane and is involved in immunity [79]. Largely due to the insufficient compatibility of using first- and second-generation PBLs of bacterial origin in plants, it was only with the introduction of the third-generation enzymes such as Turbo and miniTurbo, working optimally at temperatures far below 37 °C, that PBL became more proficient in identifying PPIs *in planta* [80,81,82], making BioID amenable to boost bacterial effector biology research in the context of plant hosts in the future.

An alternative proximity labeling (PL) approach, relies of the translational fusion of engineered APEX to biotinylate proteins in the vicinity of the bait [85]. In the presence of H_2_O_2_, APEX catalyzes the oxidation of biotin-phenol to the highly reactive and short-lived biotin-phenoxyl radical that subsequently reacts with electron-rich amino acids (>98% tyrosine and <2% tryptophan, cysteine [86]) of proximal proteins in a range of ~20 nm [86], resulting in their biotinylation. As for third-generation PBLs, a more efficient version of the enzyme was generated, *i.e.*, APEX2, using directed evolution [87]. APEX(2)-based proximity labeling greatly excels in terms of labeling time compared to BioID/BioID2/BASU (1 min or less vs. >16 h) and thus even the labeling kinetics of the recently realized TurboID and miniTurbo PBLs (~10 min of labeling). APEX2 has enabled the study of time-sensitive protein complexation, such as G-protein-coupled receptor signaling [88,89]. Because of its toxicity, the use of H_2_O_2_ however warrants caution and its use is unsuitable in case of high endogenous peroxidase activity, therefore making TurboID and miniturbo the strategies of choice for non-toxic proximity labeling [80,90].

Overall, PL has already proven to be a potent tool to uncover eukaryotic PPI networks [91,92], next to defining subcellular or localized protein compositions [93]. In bacterial cells, proximity labeling was only implemented very recently using the advantageous kinetics of APEX2-dependent biotinylation thereby revealing proteins involved in biogenesis of the type VI secretion system (T6SS) in *E. coli* [94]. By stalling the T6SS assembly pathway at different stages, Santin and co-workers were able to pinpoint the consecutive proximal partners of TssA during T6SS assembly. The first report on EH-PPIs mapping using BioID was in a study aiming at the identification of host targets of the newly discovered *Chlamydia psittaci* (*C. psittaci*) T3E SINC [84]. BioID revealed that SINC targeted the host nuclear envelope, thereby contributing potential vital knowledge of *C. psittaci* virulence. More recently, BioID was successfully applied as a large-scale screening method for the discovery of *Salmonella* T3E-host protein interactions [53]. D’Costa and colleagues screened the *Salmonella* effectors SopD2, SifA, PipB2, SseF and SseG for human protein interactors using inducible BioID effector fusions expression in stable HEK293T epithelial cell lines. The presence of an additional FLAG-tag enabled the application of BioID and AP-MS in parallel. Using BioID, eight known besides 632 putative novel (indirect) EH-PPIs were identified, the latter category showing a significant enrichment of membrane proteins and proteins implicated in vesicle organization-, cytoskeleton-, and endosomal transport when compared to AP-MS data. Five of the newly discovered candidate EH-PPIs were validated through co-IP in the case of SifA, including the interaction with the host multi-subunit BLOC-2 complex implicated in lysosomal trafficking, a complex not picked up by traditional AP-MS. In addition, since bacterial effectors notoriously target biological membrane (-associated) complexes [95], this study underscores the utility of BioID and PL in general in elucidating bacterial EH-PPIs and further illustrates the value of robust high-throughput screening platforms in increasing our understanding of effector biology.

### 2.3. Virotrap

By allowing stringent IP conditions or the direct identification of biotinylated sites [96], BioID benefits from the covalent biotin modification to circumvent false positive or negative PPIs due to cell lysis and nonspecific interactions [87]. Conversely, Virotrap eliminates the need for cell lysis altogether by making use of the vesicle-forming properties of the human immunodeficiency virus type 1 (HIV-1) GAG protein (55 kDa) [97]. Expression of an N-terminally GAG-tagged bait results in the budding of so-called virus-like particles (VLPs)—wherein interacting proteins are “trapped”. Important for downstream purification of these VLPs, is the co-expression of vesicular stomatitis virus G protein (VSV-G) in both the FLAG-tagged and untagged form (i.e., expression of both forms facilitate trimerization). VSV-G(-FLAG) appears as a composite trimer at the cell membrane and consequently, after budding, makes up part of the VLP surface. As a result, VLPs can simply be isolated from the growth medium using immobilized anti-FLAG antibodies. Next, VLP contents are liberated and typically analyzed using label-free shotgun proteome analysis. It should be noted that natural GAG protein is thought to reside in the cytosol before multimerization at the plasma membrane, at multivesicular bodies or at similar surfaces [98]. Hence, the endogenous location of Virotrap bait-interactions should preferentially be the cytosol, as this is the most physiologically relevant location for interactions in Virotrap. Currently, studies on 33 Virotrap target hits [98,99] that reside in or at the plasma membrane or fulfill (one of) their function(s) in the cytosol have been published.

To date, Virotrap has only been optimized in HEK293T cells. Nonetheless, with certain optimizations, Virotrap would likely be applicable to other mammalian cells and can thus potentially be extrapolated to other species. In this context, it is noteworthy that VLPs have successfully been isolated from plant tissue expressing full-length HIV-1 GAG [100], so application of Virotrap in plants could be an interesting subject of future investigation. Virotrap has fruitfully assisted in selecting candidate interactions for human ring finger protein 41 (RNF41) and proved to be orthogonal to BioID and AP-MS in that regard [97,99]. Furthermore, we recently obtained proof-of-concept data that Virotrap can be used for studying EH-PPIs by fusing the C-terminus of GAG to *Salmonella* T3Es, by the identification of known host interactors (unpublished data). With respect to its complementarity to other high-throughput methods, we anticipate that Virotrap will further aid the comprehensive discovery of EH-PPI maps.

## 3. Discussion

The availability of various interactomics approaches and their application have provided insights into the strengths and weaknesses with respect to studying host-pathogen interactions. The often complementary nature of the different interactomics methods currently at hand, stresses the need to use more than one approach to obtain a complete and less biased picture of EH-PPI repertoires (Figure 3), as illustrated by several studies discussed above [99,101,102]. AP-MS and derived technologies thereof are usually not successful in detecting weak and dynamic PPIs, due to their highly dynamic nature, or membrane targets, due to their poor solubilization and thus extraction during native protein extraction.

The development of techniques such as BioID and Virotrap (partially) alleviated these shortcomings by means of covalent modification and avoiding cell lysis, respectively [91,97]. More specifically, BioID on the one hand takes advantage of marking PPIs by means of the covalent attachment of biotin, a stable modification that is retained during cell lysis conditions, opposed to the necessity for the maintenance of interactions upon cell lysis in case of AP-MS analysis. Virotrap on the other hand benefits from its unique feature being the evasion of cell lysis by hijacking vesicle-forming properties of HIV-1 GAG protein. In addition, especially for poorly soluble proteins, such as membrane and cytoskeletal proteins, BioID and Virotrap have proven to clearly outperform AP-MS screenings [97,99,101,102]. Since the total amount of membrane-associated effectors is estimated to be around 30%, there is a clear necessity for techniques capable of targeting this protein category [95]. BioID takes the identification of PPIs a step further by identifying the so-called “proxeome”, inevitably however making selection of direct effector–host targets relevant for effector function a more laborious task. Selection of appropriate controls and linkers may facilitate and streamline downstream validation efforts [72,103]. GFP fused to a PBL, for example, has frequently been used as a control for non-bait specific biotinylation [83]. In addition, Virotrap has proven to be complementary to BioID in identifying PPIs [99], and both techniques are based on a completely different principle, making the methods also intrinsically complementary. Although both techniques deliver only partially overlapping protein identifications, they also harbor a specific set of missing interactions, or false negatives. For instance, proteins lacking an (accessible) lysine cannot be biotinylated and will not be perceived in case of BioID. Proteins that are too distant from the BioID tag will also be missed. The latter can potentially be overcome by incorporating a (longer) linker between the tag and the bait, which, in addition, reduces the effects of steric hindrance (Figure 4) [73], or alternatively, by re-positioning of the PBL. False negative interactions in Virotrap might include proteins residing in confined subcellular structures of organelles, such as the nucleus, or protein complexes too large to fit in the VLP particles estimated to be 100 to 350 nm in diameter [104]. False positives can to a certain extent be limited by an appropriate experimental design. Endogenously biotinylated proteins, for instance, can easily be filtered out by comparison with a control dataset representing the background. False positive interactions with GAG can be eliminated by comparing to a control setup. An appropriate VLP control setup, i.e., a representation of the complete VLP proteome in absence of the effector, is indispensable for Virotrap MS-based analysis as VLPs exhibit their own characteristic proteome content [97]. In the context of appropriate BioID controls, it is noteworthy that by using the power of genome editing, Vandemoortele et al. designed a method to obtain isogenic cell lines expressing T2A-BirA* bait fusion or its corresponding BirA* control protein, both under the control of the endogenous bait promoter [105]. The method relies on introduction of the T2A-BirA* cassette at the C-terminus of the protein of interest resulting in bicistronic expression of both the bait and BirA*, thereby serving an elegant control setup for isogenic cells obtained by targeting of a Cas9 enzyme, fused to a cytidine deaminase to the T2A autocleavage site, thereby inactivating the T2A peptide sequence resulting in the expression of the bait fusion from its endogenous promoter. It is clear that EH-PPI discovery is multifaceted and requires careful assessment of the effector protein under investigation to set up a successful experiment. The study of effector biology is further complicated by the range of susceptible hosts. Therefore, selection of an appropriate host context is of essential importance, also viewing the reported overreliance on a few popular models clearly biasing host-pathogen interactomics studies [106]. We recently demonstrated that EH-PPIs of bacterial root pathogens can be studied when making use of hairy root cultures closely mimicking the environment of the natural host [107] (unpublished data). In this setup, roots are transformed with *Agrobacterium rhizogenes* carrying the desired construct of for example a tagged effector. Transformed roots can be selected for (e.g., by co-expression of a fluorescent marker) and used as model for protein interaction studies [83]. It is also important to consider the mode of effector tagging. Of note, signal peptides, CBDs and for instance GTP-activating domains typically reside at the N-terminus of bacterial effectors [108]. The latter suggests a general preference for C-terminally tagged effector fusions to retain functionality, but it is always essential to validate functionality of the tagged variant independent of tag size.

Investigation of bacterial effectors in their physiological context can be essential for proper effector functioning. This is for instance demonstrated by the observed aberrant localization of certain heterologously expressed effectors versus the localization of effectors delivered through the bacterial secretion system by infection of host cells, whether or not following endogenous chromosomal tagging [109,110] or overexpression in bacteria [111]. Illustrative of this, while SifB is known to be targeted to the *Salmonella*-containing vacuole (SCV) and *Salmonella*-induced filaments (Sifs) when secreted by *Salmonella*, i.e., when SifB is present at physiological levels, a cytoplasmic localization was observed upon heterologous (over-)expression [112]. Of note, the lack of EH-PPI identifications for some effectors studied by Sontag et al. could be due to improper effector production or localization, or the absence of specific cofactors or protein modifications in vitro. Moreover, many effectors rely on protein modifications or the expression of other effectors on which their function relies. EPEC Tir (translocated intimin receptor) phosphorylation (Y454), for example, leads to recruitment of N-WASP, resulting in actin assembly at the site of bacterial invasion [113]. The T3E SopD from *S*Tm relies on the phosphatase activity of a second T3E SopB for its targeting to the plasma membrane [114]. As such, effectors are preferably studied upon native delivery in infection(-relevant) conditions. One way is to make use of bacteria encoding endogenously tagged effectors when infecting host cells. A potential strategy based on BioID would be to tag effectors with a PBL at their endogenous locus (endogenous BioID, eBioID, Figure 4). Secretion of tagged T3Es and T4Es have been proven successful in case of small tags and certain reporters [109,111,115]. However, viewing the limited unfolding capacity of the T3SS (or T4SS [116]) and the stable β-barrel structure of fluorescent proteins (e.g., GFP), fusion of fluorescent proteins to T3Es was shown to block T3SS-mediated secretion [13], thereby generally excluding the use of tightly folded proteins as T3E fusions. However, a split GFP system was developed that enables GFP-IP and real-time imaging of effector translocation. By tagging the effector with strand 11 from GFP and expressing the other ten strands in the desired host cell, delivery, and localization of, for instance, three *S*Tm effectors (i.e., PipB2, SteA and SteC) [109] and two *R. solanacearum* GMI1000 effectors (i.e., AvrB and PopP2) was successfully accomplished [117]. Alternatively, other modules with likely higher unfolding capacity as compared to GFP, such as PhoA, calmodulin-activated adenylate cyclase and beta-lactamase, have been successfully used to monitor effector secretion [118,119,120]. Upon native delivery of tagged effectors, and to improve the resolution of future host-pathogen interactomics studies, it will be increasingly important to additionally consider spatiotemporal effector expression profiles and kinetics of effector delivery. As illustrated by the Knodler et al. study profiling T3E-host PPIs at later stages of infection (i.e., 20 hpi), a bias towards the identification of *Salmonella* SPI-2 T3Es host targets compared to SPI-1 targets could clearly be observed. This discrepancy is largely due to the fact that SPI-2 T3Es are translocated at later stages of infection when bacterial loads are high, implying their generally higher abundance and concomitant improved detectability of SPI-2 host targets [56]. The implementation of high sensitive data-independent acquisition (DIA)-based proteomics approaches shown to outperform data-dependent acquisition (DDA) workflows in identifying and quantifying low abundant proteins, may aid future EH-PPI studies of low abundant effectors [121]. All considering, it should be attempted to closely mimic real infection conditions, but a delicate balance between accuracy and feasibility is ultimately essential.

Creative approaches using the combined power of AP-MS and BioID have enabled diversion of relative spatial distances for proteins within a complex [101,122]. Liu et al. introduced this principle using a MAC-tag (StrepIII-BirA*), enabling the integration of stoichiometric complexation information from AP-MS with the identification of transient or proximal interactions by BioID. D’Costa et al. also used a composite tag, but used it to perform both methods in parallel (enabling the comparison of identical (tagged) baits besides reducing cloning efforts) [53].

Furthermore, it is important to consider that effectors with enzymatic activity might not form stable interactions with their substrates and that protein interactions with other macromolecules, such as lipids or DNA/RNA, should not be disregarded when considering pathogenesis mechanisms. Substitution of the described methodologies by other technologies to frame these interactions is thus desirable. Interestingly, BASU was recently reported to capture RNA-protein interactions using labeling times as short as 1 min [123].

Implementing complementary MS-based strategies was proven very useful in distinguishing background from relevant PPIs. Integration of these rich host-pathogen interactomics datasets with knowledge of timing of effector expression, secretion, activity and (subcellular) localization will eventually enable us to shed better lights on bacterial pathogenesis.

## Figures and Tables

**Figure 1 ijms-21-06891-f001:**
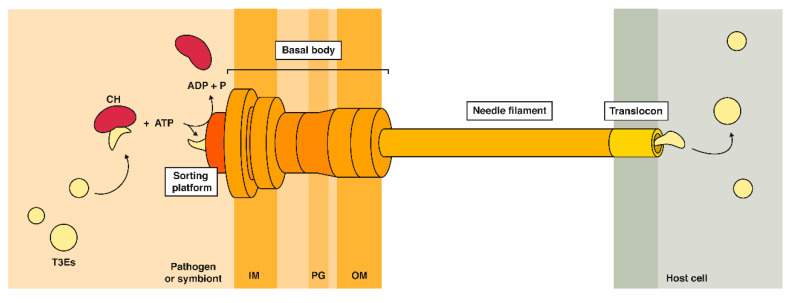
The type-III secretion system delivers bacterial effectors into the host cell by acting as a molecular syringe. Chaperones (CHs) are believed to keep type-III effectors (T3Es) in a partially unfolded state inside the bacterium. CHs are released from the T3Es at the sorting platform and are brought into the hollow channel of the needle filament for subsequent release inside the host cell cytoplasm. IM = inner membrane; PG = peptidoglycan; OM = outer membrane.

**Figure 2 ijms-21-06891-f002:**
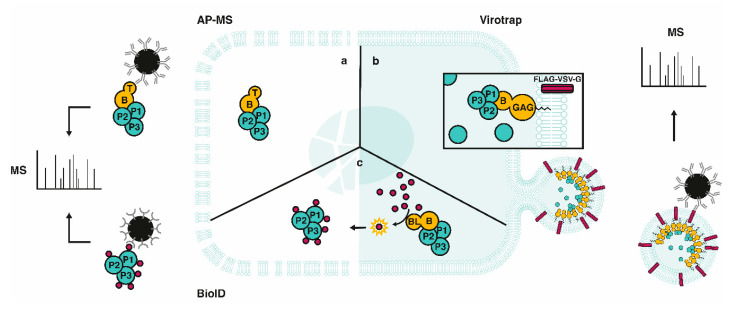
Complementary MS-based co-complex platforms to capture protein–protein interactions. (**a**) AP-MS; In AP-MS, the bait protein (B) is fused to an epitope tag (T) that has a high affinity to an immobilized antibody. The bait interactome (P1-P3) is co-purified after native cell lysis and subsequently identified through mass spectrometry (MS). (**b**) Virotrap; Virotrap omits the need for cell lysis by the genetic fusion of a bait to the myristoylated (zigzag line) human immunodeficiency virus type 1 (HIV-1) GAG protein. Expression of the fusion protein elicits the aggregation of the GAG portion at the plasma membrane, enabling the budding of virus-like particles (VLPs) and “trapping” of host preys inside VLPs. Anti-FLAG purification of the VLPs in the culture medium is followed by MS-based analysis of the VLP content (**c**). BioID; In BioID, the genetic fusion of a protein biotin ligase (BL) to a protein of interest, or bait (B), results in the in vivo biotinylation of interacting prey (P) proteins that are captured after lysis via streptavidin-based purification for subsequent MS-based analysis. Purple circles represent biotin.

**Figure 3 ijms-21-06891-f003:**
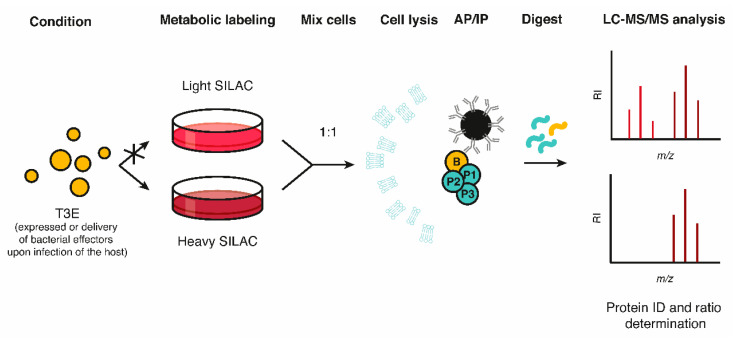
Defining protein interactions using stable isotope labeling of amino acids in cell culture (SILAC)-based quantitative mass spectrometry. In the representative example shown, SILAC comprises differential metabolic labeling (light and heavy stable isotope-labeled samples) of proteins in distinct experimental conditions (− and + effector expression/delivery, in case of the light and heavy SILAC sample, respectively), followed by mixing of the resulting proteomes and a subsequent affinity- or immunopurification (AP/IP) step targeting the effector bait (B). After digestion of the resulting mixed protein sample, relative protein abundances, and concomitantly, specific effector interactors (significantly enriched as heavy label; P1-P3) can easily be distinguished from contaminants by MS-based analysis. RI = relative intensity.

**Figure 4 ijms-21-06891-f004:**
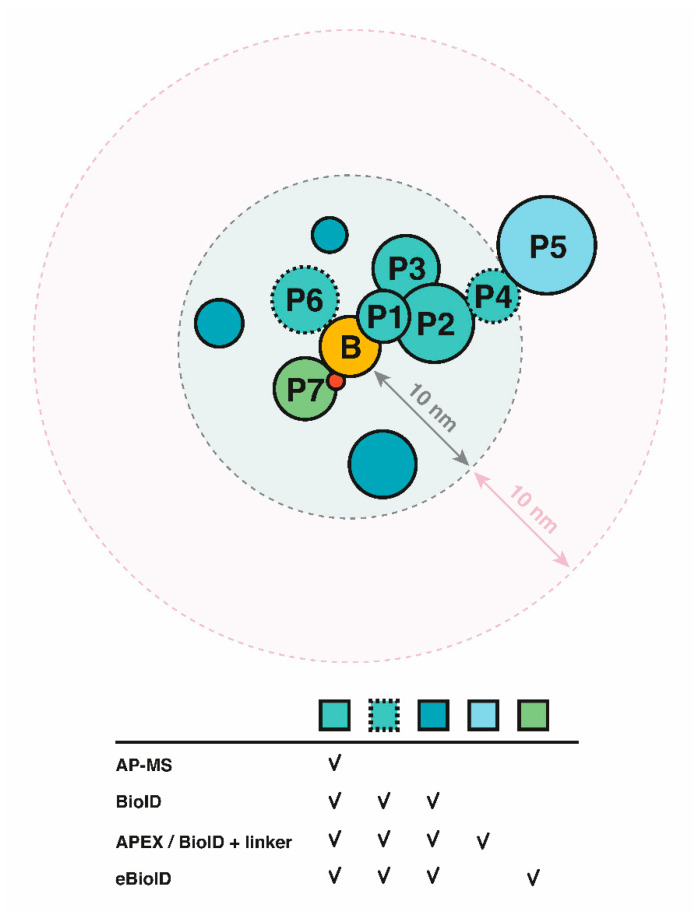
Different MS-based co-complex platforms are complementary in nature. Protein identification by AP-MS, BioID, APEX or BioID with linker and endogenous BioID (eBioID) is depicted. B = bait; P = protein interactor (direct/indirect); red and dotted circles represent post-translational modification and weak or dynamic interactors, respectively.

**Table 1 ijms-21-06891-t001:** Overview of type-III effectors host targets identified using AP-MS. Host interactors indicated with an asterisk were validated using co-immunoprecipitation. Underlined host interactors were found in multiple independent interactomics studies.

Bacterial Pathogen	Type-III Effector	(Candidate) Host Interactors (Gene Names)	Proteomics Approach	Species or Cell Type	Ref.
***Pseudomonas syringae***	HopF2	AHA2, AHA11, PDR8/PEN3, ERD4, PIP2A, PIP3, Clathrin heavy chain, ADP/ATP Carrier protein, **REM1.3**, HIR2	AP-MS	Arabidopsis	[56]
	HopM1	UPL1, UPL3, ECM29, proteins related to 26S proteasome non-ATPase regulatory subunits 2, 3, 6, 12, 14, BIG, orthologues of AtMIN7 and AtMIN10	AP-MS (GFP-trap)	Tobacco	[57]
	HopQ1	TFT1*, TFT2, TFT3, TFT4, TFT5*, TFT6, TFT7, TFT9, TFT10	AP-MS	Tomato (cv. Moneymaker)	[58]
***Ralstonia solanacearum***	RipAY	NbTRX-h11, NbTRX-h09, NbTRX-h10, NbTRX-h15*	AP-MS (GFP-trap)	Tobacco and Arabidopsis	[59]
***Chlamydia trachomatis***	58 Inc-class effectors	354 high-confident PPIs	AP-MS	HEK293T	[48]
***Citrobacter rodentium***	EspT	HSPD1	AP-MS (in vitro)	RAW 264.7 and HeLa lysates	[51]
	NleA	LDHB, PHGDH, SEC24B, DLG1, SEC23A, SLC3A2			
	NleG1	TUFM, GAPDH, UQCRC2, PKM, MCM7, PRKDC, CPS1, SLC25A6, SLC25A5, SERPINH1, PHGDH, ACADM			
	NleK	HNRNPM			
***Salmonella enterica* serovar Typhimurium**	SopB	CDC42	AP-MS (SILAC)	HEK293T	[61]
	SspH2	SUGT1*, AIP, BUB3*, YWHAG, BAG2			
	SseJ	RHOA, RHOC			
	SspH1	PKN1			
		PKN1*	AP-MS	RAW 264.7 and HeLa lysates	[51]
	SseG	DSP, CAPRIN1	AP-MS (SILAC)	HEK293T	[62]
		MYH10, IPO5, PHB2, MYL12B, EPHX1, RANBP6, EIF3B, NNT, SDHA, EIF3A, VDAC1, OCIAD1, NDUFA13, FAM162A, ARL6IP5, GK, API5, EIF3E, COX5B, VDAC2, PSMD12, RAB8A, AP3D1, AGK, CLPTM1L, CUL4B, VAMP3, BAX, CYP51A1, HMOX2, RDH11, TMEM48	AP-MS	HEK293T	[52]
	SseL	OSBPL1A*, TLN1	AP-MS (SILAC)	HEK293T	[63]
		NEDD8, TXN, PSME2, S100A6, RCC2, S100A11, PRDC1, UBA52	AP-MS	RAW 264.7 and HeLa lysates	[51]
	SseF	JUP	AP-MS (SILAC)	HEK293T	[62]
		RBM10, THRAP3, ARGLU1	AP-MS	HEK293T	[52]
	GogA	PRPF31	AP-MS	RAW 264.7 and HeLa lysates	[51]
	GtgA	MOGS, SLC25A11, PTGES2, SSR1, ATP5O, USMG5, GPNMB, BCAP29, ALDH3B1, 1700055N04RIK, RPN2, HADHA, YME1L1, ABHD12, IQGAP1, GALNT7, SGPL1, HSD17B12, CYC1, SLC25A12, SLC25A13, ACSL4, GM10250, B4GAINT1, LRRC59			
	GtgE	LYN, GOPC			
	SpvC	LPXN			
	SrfH	DNAJA1, ABCF2, ERK2*, UPF1, PFKI, MSH2, GM9755, SUCLG1, GALK1, GRPEL1, ACADM, PFKP, EPRS, IDH3B, SLC25A12			
	SssB	GRN			
	SifA	MYH10, MYL6, MYL12B, EIF3B, RBM10, HM13, EIF3A, CDIPT, EIF5AL1, AP3D1, NDUFA13, TMEM59, ATP5D	AP-MS	HEK293T	[52]
	PipB2	GCN1L1, XPO1, **IRS4**, MYH10, FANCI, XPO5, ATP2B1, FKBP5, SUGT1, NCAPD2, MYH9, SCRIB, KIAA0368, TNPO1, LRRC1, TELO2, DIS3, ACTA1, DDX19A, AP3D1, PDS5A, MMS19, HDDC2, MYL12B, CAND2, NTPCR, RHOG, TRMT1, CDC73, YTHDF2, RDH11, ANXA4, PELO, UMPS, PRPF6, NAA15, PSMD12, MTHFD1L, EEF1E1, ADCK3, PLAA, CALM2, UROD, ANP32E, FABP5, LTN1, MLF2, SYNE1, ATP6V1H, CUL1, NEDD8-MDP1, FBXO22, SNRPG, UBXN1, AUP1, PIN1, LYPLA2, ARIH1, PCID2, LARP4B, CELF1, ARGLU1, GOLPH3, ORC3, DDX23			
	SopD2	MYH10, MYL6, MYL12B, MYH9, RBM10, RAB10, EIF3B, CYFIP1, PHB2, EIF3A, AP2B1, EIF3E, AP3D1, MYO1B, RAB8A, AP3B1, AP2A1			
	SopA ^a^	TRIM56*, TRIM65*, HDAC10, GSTM3, PCMT1, MAPK3, AP2B1, XRCC5, PPP2R2A, XRCC6	AP-MS (SILAC; GFP-trap)	HeLa	[63]
		TRIM56*, TRIM65*, EPS15L1, GTF2F2, PDLIM7, CSTF1, GTF2F1, RAD23B, MAPRE1, G6PD	AP-MS (SILAC)	HCT116	
	15 *S*Tm T3Es	446 high-confident PPIs	AP-MS (delivery of chromosomally tagged T3Es in the context of infection)	HeLa and RAW 264.7	[55]
**EPEC**	Map	NERF2	AP-MS (SILAC)	HEK293T	[64]
	EspJ	WDR23			
	EspL	MAP7*			
	EspX	MAP7			
	NleA	SEC23A, SEC24B, DLG1			
	NleB1	MAP7*			
	NleC	P300			
	EspZ	CD98*, RPS27A, HSP90AB1, HSP90AA1	AP-MS (SILAC)	HEK293	[61]

^a^ only the ten most significant SopA-enriched hits are listed. Candidate host interactors indicated in bold were also found using BioID (see Table 2).

**Table 2 ijms-21-06891-t002:** Overview of type-III effectors host targets identified using proximity labeling. Host interactors indicated with an asterisk were validated using co-immunoprecipitation.

Bacterial Pathogen	Type-III Effector	(Candidate) Host Interactors (Gene Names)	Proteomics Approach	Species or Cell Type	Ref.
***Pseudomonas syringae***	HopF2	19 ^a^ RIN4, **REM1.3**, PCAP1, PHOT1, PHOT2, SYP122, PMI1, PATL1, PATL2, RBCS1A	BirA*	Arabidopsis	[77]
	AvrPto	25 ^a^ RIN4, APK1, APK2, TOM1, APP4*	BirA*	Tobacco	[79]
***Chlamydia psittaci***	SINC	22 ^a^ ELYS, laminB1*, emerin*, MAN1, LAP1, LBR	BirA*	HeLa	[83]
	IncF	13 ^a^ LRRF1, MAP1B, CYTB, BASP1, YWHAH, MARCKS, YWHAB, K1C20	APEX2	HeLa	[84]
	IncA_TM_	18 ^a^ LRRF1, MAP1B, CYTB, BASP1, MYPT1, TERA, MAP4, PUR6, SNX1, SRC8, MEP50, SNX6, PLIN3, IF4B, NDKA, NDKB			
	IncA	192 ^a^ LRRF1, MAP1B, CYTB, BASP1, MYPT1, TERA, MAP4, PUR6, SNX1, SRC8, MEP50, SNX6, PLIN3, IF4B, NDKA, NDKB, YWHAH, MARCKS, YWHAB, K1C20			
***Salmonella enterica serovar*** **Typhimurium**	SifA	167 ^b^ BLOC-2*, IRS4, EPB41L3, EPB41L2, MLLT4, DST, LBR, YKT6, TMPO, SLC12A2	BirA*	HEK293T	[52]
	PipB2	149 ^b^ KIF5B, EPB41L3, **IRS4**, TMPO, LEMD3, EPB41, MLLT4, KLC1, EPB41L2, KLC2			
	SseF	107 ^b^ TUFM, IRS4, LBR, EPHA2EPB41L2, UBB, DDX17, CKAP4, PTPLAD1, HNRNPC			
	SseG	145 ^b^ TMPO, ESYT1, LBR, PGRMC2, LEMD3, CKAP4, ABCD3, RPN1, ACSL3, LMAN1			
	SopD2	61 ^b^ RUVBL1, RUVBL2, HSP90AA1, CCT2, CLCN7, SUGT1, RAB7A, YKT6, ZFYVE16, HNRNPAB			

^a^ hits listed are top candidates selected by the researchers. ^b^ only the ten most significantly enriched hits are listed. Candidate host interactors indicated in bold were also found using AP-MS (see Table 1).

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
