# Peer review of "Keeping in Touch with Type-III Secretion System Effectors: Mass Spectrometry-Based Proteomics to Study Effector–Host Protein–Protein Interactions"

_ijms, 2020, doi:10.3390/ijms21186891_

Round 1

Reviewer 1 Report

The authors De Meyer et. al. have written a well-organised, excellent review on the current proteomic platforms/methods available to study and identify effector-host protein interactions. The review also highlights emerging methods that can improve on the efficacy of those currently used. The literature is fairly and thoroughly discussed, with limitations and challenges of methods covered. The figures illustrate and summarise the platforms discussed well.

The authors could consider providing sub-headings within the Introduction to break up the text, particularly when discussing the function of T3Es etc.

Only a few grammatical errors/suggestions:

Line 41- has on its turn- has in turn

Line 65- evolutionarily related

Lines 83-84- consider improved punctuation or rephrasing?

Line 136- in the case of

Line 138- 'and for mapping'- information missing?

Line 171- followed by MS

Consider splitting figure 1 into 2 figures, with BioID and Virotrap illustrated later in section 2

Figure 1 legend- 'budding off of' change to 'budding of'

Line 299- consider using nonspecific over unspecific

Line 369- in the case of

Line 402- 'Viewing its complementarity'- please consider rephrasing

Line 416- Figure 3 legend - identification by?

Line 438- in the case of

Line 454- serving as an

Line 527- shed better light (singular)

Author Response

Response to Reviewer 1 Comments

The authors De Meyer et. al. have written a well-organised, excellent review on the current proteomic platforms/methods available to study and identify effector-host protein interactions. The review also highlights emerging methods that can improve on the efficacy of those currently used. The literature is fairly and thoroughly discussed, with limitations and challenges of methods covered. The figures illustrate and summarise the platforms discussed well.

The authors could consider providing sub-headings within the Introduction to break up the text, particularly when discussing the function of T3Es etc.

We would like to thank the reviewer for his kind appreciation of our work. As suggested by the reviewer we now provided extra subheadings in the introduction.

Only a few grammatical errors/suggestions:

Line 41- has on its turn- has in turn

Line 65- evolutionarily related

Lines 83-84- consider improved punctuation or rephrasing?

Line 136- in the case of

Line 138- 'and for mapping'- information missing?

Line 171- followed by MS

Consider splitting figure 1 into 2 figures, with BioID and Virotrap illustrated later in section 2

Figure 1 legend- 'budding off of' change to 'budding of'

Line 299- consider using nonspecific over unspecific

Line 369- in the case of

Line 402- 'Viewing its complementarity'- please consider rephrasing

Line 416- Figure 3 legend - identification by?

Line 438- in the case of

Line 454- serving as an

Line 527- shed better light (singular)

All grammatical errors were adjusted accordingly. We however preferred to keep the layout of our original figure 1 (i.e., now figure 2, in the revised version of the manuscript) as we believe it provides a good overview/illustration of the main techniques discussed in this review.

Reviewer 2 Report

This is a fairly straightforward review, well focused and updated. I have no major problems with it. I’m suggesting a few points that the authors might wish (or not) to consider.

The Ms presented by De Meyer and coworkers is a review on the on MS- based proteomics approaches to study the interaction between host and Type III secretions system effector proteins. The review is clear and well-constructed and contains a wealth of updated information. On the overall is worth publishing. I’d like only to remark a few issues that the authors could consider to perhaps improve their work.

1.- Some sentences are a bit difficult to understand. Examples are:

L. 33-34 “These elicitors are termed either microbial (microbe-associated molecular patterns or MAMPs) or pathogenic (pathogen-associated molecular patterns or PAMPs).”. Wouldn’t be better: “These elicitors are termed either microbe-associated molecular patterns (MAMPs) or pathogen-associated molecular patterns (PAMPs).”?

L79. “As opposed to the highly conserved T3SS machinery [9], its translocated type III effectors are..”. Consider “As opposed to the highly conserved T3SS machinery [9], type III effectors are..”.

L. 264. “upon mammalian host cell T3E expression or recombinant addition of the T3E…”. Use instead, “upon mammalian host cell T3E expression or addition of recombinant T3E..”

2.- I find a bit confusing the difference between AP-MS and IP-MS as described in the text. The examples for AP do use antibodies also.

3.- Although I understand that this is not the main aim of the review, the authors dedicate some effort in the introduction to describe the structure of type III secretion systems. I wonder if addition of a cartoon depicting such structure could help to better understand it.

4.- REM1.3 is not in bold in Table 1, and I believe it should be.

5.- Not being an expert in type III ss and T3E, I’m surprised that virtually nothing is said about chemical cross-linking followed by MS. There are no examples of such approach in this field?

Author Response

Response to Reviewer 2 Comments

This is a fairly straightforward review, well focused and updated. I have no major problems with it. I’m suggesting a few points that the authors might wish (or not) to consider.

We again would like to thank the reviewer the appreciation of our work.

The Ms presented by De Meyer and coworkers is a review on the on MS- based proteomics approaches to study the interaction between host and Type III secretions system effector proteins. The review is clear and well-constructed and contains a wealth of updated information. On the overall is worth publishing. I’d like only to remark a few issues that the authors could consider to perhaps improve their work.

1.- Some sentences are a bit difficult to understand. Examples are:

  1. 33-34 “These elicitors are termed either microbial (microbe-associated molecular patterns or MAMPs) or pathogenic (pathogen-associated molecular patterns or PAMPs).”. Wouldn’t be better: “These elicitors are termed either microbe-associated molecular patterns (MAMPs) or pathogen-associated molecular patterns (PAMPs).”?

We would like to thank the reviewer for noticing this overtly complex sentence. This sentence was now revised accordingly.

L79. “As opposed to the highly conserved T3SS machinery [9], its translocated type III effectors are..”. Consider “As opposed to the highly conserved T3SS machinery [9], type III effectors are..”.

We adjusted the sentence accordingly.

  1. 264. “upon mammalian host cell T3E expression or recombinant addition of the T3E…”. Use instead, “upon mammalian host cell T3E expression or addition of recombinant T3E..”

Adjusted.

2.- I find a bit confusing the difference between AP-MS and IP-MS as described in the text. The examples for AP do use antibodies also.

We now made it clearer in the text that in case of IP-MS co-purification of interactors is performed using antibodies specific for the selected (untagged) bait protein, while in case of AP-MS, co-purification of tagged bait interactors is performed by performing an IP with tag-specific antibodies.

3.- Although I understand that this is not the main aim of the review, the authors dedicate some effort in the introduction to describe the structure of type III secretion systems. I wonder if addition of a cartoon depicting such structure could help to better understand it.

As suggested by the reviewer, a new illustrative figure (i.e., new figure 1) depicting the type III secretion system as a molecular syringe to deliver effectors into the host cell has now additionally been included.

4.- REM1.3 is not in bold in Table 1, and I believe it should be.

We would like to thank the reviewer for this important note and now indicated REM1.3 in bold type interface, as REM1.3 was indeed identified as an HopF2 interactor using AP-MS as well as BioID.

5.- Not being an expert in type III ss and T3E, I’m surprised that virtually nothing is said about chemical cross-linking followed by MS. There are no examples of such approach in this field?

Besides referencing/discussing a study which made use of chemical crosslinking, we now included some extra sentences on the use of chemical crosslinking when studying HP-PPIs (lines 184-188 and lines 269-271).